# Dental Stem Cells and Lipopolysaccharides: A Concise Review

**DOI:** 10.3390/ijms25084338

**Published:** 2024-04-14

**Authors:** Beatriz A. Rodas-Junco, Sandra E. Hernández-Solís, Angelica A. Serralta-Interian, Florencio Rueda-Gordillo

**Affiliations:** 1CONAHCYT–Facultad de Ingeniería Química, Campus de Ciencias Exactas e Ingenierías, Universidad Autónoma de Yucatán, Periférico Norte Kilómetro 33.5, Tablaje Catastral 13615 Chuburná de Hidalgo Inn, Mérida CP 97203, Yucatán, Mexico; 2Laboratorio Traslacional de Células Troncales de la Cavidad Bucal, Facultad de Odontología, Universidad Autónoma de Yucatán, Calle 61-A #492-A X Av. Itzaes Costado Sur “Parque de la Paz”, Col. Centro, Mérida CP 97000, Yucatán, Mexico; angelicaserralta@gmail.com; 3Departamento de Microbiología Oral y Biología Molecular, Facultad de Odontología, Universidad Autónoma de Yucatán, Calle 61-A #492-A X Av. Itzaes Costado Sur “Parque de la Paz”, Col. Centro, Mérida CP 97000, Yucatán, Mexico; hsolis@correo.uady.mx (S.E.H.-S.); gordillo@correo.uady.mx (F.R.-G.); 4Facultad de Ingeniería Química, Campus de Ciencias Exactas e Ingenierías, Universidad Autónoma de Yucatán, Periférico Norte Kilómetro 33.5, Tablaje Catastral 13615 Chuburná de Hidalgo Inn, Mérida CP 97203, Yucatán, Mexico

**Keywords:** lipopolysaccharides, dental stem cells, periodontal disease, pulp inflammation

## Abstract

Dental tissue stem cells (DTSCs) are well known for their multipotent capacity and regenerative potential. They also play an important role in the immune response of inflammatory processes derived from caries lesions, periodontitis, and gingivitis. These oral diseases are triggered by toxins known as lipopolysaccharides (LPS) produced by gram-negative bacteria. LPS present molecular patterns associated with pathogens and are recognized by Toll-like receptors (TLRs) in dental stem cells. In this review, we describe the effect of LPS on the biological behavior of DTSCs. We also focus on the molecular sensors, signaling pathways, and emerging players participating in the interaction of DTSCs with lipopolysaccharides. Although the scientific advances generated provide an understanding of the immunomodulatory potential of DTSCs, there are still new reflections to explore with regard to their clinical application in the treatment of oral inflammatory diseases.

## 1. Introduction

In humans, the oral cavity contains a diverse ecosystem of microorganisms, including viruses, fungi, archaea, and bacteria [1], that positively or negatively interact with their host. Inflammatory diseases caused by bacteria (both gram-positive and gram-negative) create an inflammatory microenvironment that encourages the formation of an oral biofilm, which can result in conditions such as pulpitis, gingivitis, and periodontal diseases [2,3]. This affects the biological characteristics of dental tissue stem cells (DTSCs), including their ability to control these infections. DTSCs are a readily available source of autologous mesenchymal cells. They also have differentiation potential toward various cell lineages and possess strong immunomodulatory functions [4,5]. DTSCs are also targeted by inflammatory agents such as bacterial endotoxin (lipopolysaccharide), which leads to the release of paracrine substances such as chemokines [6,7,8]. Lipopolysaccharides (LPS) are pathogen-associated molecular patterns found in the cell walls of gram-negative bacteria that can trigger an immune response in the host cell [9,10]. Structurally, LPS have variable and conserved regions [11,12], and any alteration in their chemical configuration will affect the overall structure of the molecule [13]. Although it has been reported that dental stem cells use cell surface receptors to recognize various LPS species [14], the molecular mechanisms involved in an inflammatory microenvironment and the commitment to differentiation in dental stem cells remain poorly understood. This review aims to provide insight into the biological behavior of DTSCs during their interaction with bacterial lipopolysaccharides. Additionally, we summarize the latest representative studies that highlight the cellular and molecular plasticity of DTSCs in an inflammatory environment and the challenges they face in being used as regenerative components in oral diseases.

## 2. Dental Tissue Stem Cells

Dental tissue stem cells (DTSCs) are post-natal cells; they are a significant source of mesenchymal stem cells (MSCs) that can be easily obtained through minimally invasive procedures and have been utilized in treating various diseases [15]. DTSCs exhibit high proliferation rates, the ability to differentiate into multiple lineages, and the expression of embryonic markers of MSCs such as OCT4, NANOG, SOX2, and KLF4 [4]. Different types of DTSCs exist in the oral cavity, including dental pulp stem cells (DPSCs), exfoliated deciduous tooth stem cells (SHEDs), periodontal ligament stem cells (PDLSCs), dental follicle stem cells (DFSCs), and stem cells from the apical papilla (SCAP; Figure 1). DTSCs play a crucial role in maintaining regenerative capacity. For example, DPSCs are important in the regeneration of the dentin-pulp complex [16], while PDLSCs are an important part of periodontal tissue [17]. The interest in studying DTSCs stems from their potential use in cell therapies. Understanding how these cells respond to various stimuli, including lipopolysaccharides (LPS), is essential. LPS are used in research to model an inflammatory environment in vitro in stem cells, allowing researchers to study the immunoregulatory (cytokine secretion) and regenerative effects of DTSCs in response to various bacterial toxins. The dental pulp (DP) or periodontal ligament is considered an attractive source of DTSCs due to its role in maintaining homeostasis, angiogenesis, and the vitality of dental tissue in the presence of bacterial lesions [16,18]. Therefore, understanding the molecular mechanisms associated with oral diseases such as pulpitis, periodontitis, and DTSCs can contribute to developing novel autologous treatments. In this review, we examine the current understanding of LPS derived from different bacteria and their effects on the biological characteristics of DTSCs. Finally, we discuss the molecular mechanisms involved in the interaction between LPS and DTSCs.

## 3. Lipopolysaccharides as a Component of Gram-Negative Bacteria

### 3.1. Structure of LPS Bacteria

LPS are antigenic macromolecules found in the outer membrane of gram-negative bacteria (Figure 2). LPS consist of three essential chemistry regions: the O-specific polysaccharide region (region I), the core polysaccharide component (region II), and a hydrophobic component known as lipid A (region III; Figure 1). Region I is the outermost layer of LPS and varies between species in terms of sugars, sequence, chemical linkage, substitution, and ring forms utilized, potentially containing more than 50 different repeating oligosaccharides composed of two to eight monosaccharide components. In contrast, region II has less chemical variation and consists of sequences of keto-deoxy-octulosonic acid (KDO), heptose, phosphate, and ethanolamine. KDO plays a crucial role in the ketosidic union with lipid A, the component responsible for the endotoxic activity of LPS [19]. Lipid A typically comprises a bis-phosphorylated β-1-6-linked glucosamine disaccharide that can be substituted with various groups such as phosphate, ethanolamine, phosphate ethanolamine, ethanolamine diphosphate, GlcN, 4-amino-4-deoxy-L-arabinopyranose, and D-arabinose-furanose [20].

Interestingly, the pathogenicity of LPS derived from different gram-negative bacteria is attributed to the fatty acid chain length in the lipid A structure. For example, *P. gingivalis* (a bacterium responsible for chronic periodontitis) produces different species of lipid A that can undergo acylation [tetra-(LPS_1435/1449_) and penta-(LPS_1690_) acylated], resulting in varying inflammatory responses [21]. Herath et al. (2013) observed differential expression of matrix metalloproteinases (MMP) 3 (which are synthesized in periodontal tissues in response to periodontopathic bacteria; MMP3) in human gingival fibroblasts in response to LPS_1435/1449_ and LPS_1690_ [22]. For a more in-depth exploration of the structure and function of LPS in various bacteria, we recommend referring to excellent previous reviews [11,23,24,25].

### 3.2. Lipopolysaccharides on Oral Diseases

Lipopolysaccharides act as endotoxins that can trigger septic shock as well as lead to the development of several dental diseases (Figure 1) [26]. One of the primary dental diseases in which LPS plays a role is periodontal disease, encompassing periodontitis and gingivitis [27]. In cases of periodontal disease, bacteria found in dental plaque form biofilms on teeth and gums. The release of LPS by these bacteria can initiate an inflammatory response in the gingival tissue, resulting in the breakdown of the connective tissue and bone supporting the teeth [28].

This can manifest as symptoms such as gum inflammation, gingival bleeding, gingival recession, and, in severe cases, tooth loss [29]. As the disease progresses, elevated levels of pro-inflammatory cytokines such as interleukins (IL-1 beta, IL-2, IL-6, and IL-8) and C-reactive protein are detected [30].

In the development of gingivitis, LPS plays a vital role by initiating inflammatory reactions. This process involves the stimulation of inflammatory cytokines and chemokines, including tumor necrosis factor-alpha (TNF-α), interleukin-1beta (IL-1β), interleukin-6 (IL-6), and macrophage inflammatory protein-1alpha (MIP-1α) [31], contributing to gingival tissue damage, osteoclast formation, and subsequent alveolar bone loss [32].

In periodontitis, an advanced stage of periodontal disease, an inflammatory response affects the tissues supporting the teeth, leading to the degradation of connective tissue, collagen within the gum, loss of the periodontal ligament, and resorption of alveolar bone [33]. LPS functions by triggering the production of inflammatory mediators and cytokines, subsequently stimulating the release of matrix metalloproteinases (MMPs). These tissue-derived enzymes are involved in restructuring the extracellular matrix and deteriorating bone tissue [34]. Consequently, tooth roots become exposed to the oral environment, allowing bacterial biofilm colonization on the root and root cementum, which can calcify to form dental calculus. The gradual advancement of this condition results in tooth mobility, impaired chewing ability, changes in appearance, and, ultimately, if left untreated, tooth loss [35].

In addition to periodontal disease, LPS are also implicated in dental caries. Although caries are primarily caused by acids produced by bacteria in dental plaque, LPS can contribute to inflammation and dental tissue damage if an excessive immune response occurs [36]. Moreover, the impact of LPS in the oral cavity may extend to influence overall health, potentially contributing to systemic illnesses. LPS can enter the bloodstream through the oral cavity during invasive dental procedures or due to periodontal disease. This entry into the bloodstream results in endotoxemia, correlating with increased susceptibility to systemic cardiometabolic conditions such as obesity, insulin resistance, type 2 diabetes, nonalcoholic fatty liver disease, metabolic syndrome, and dyslipidemia [37].

Furthermore, LPS may trigger a systemic immune response that may contribute to the development of arthritis, an inflammatory joint condition commonly associated with previous bacterial infections in other parts of the body. Research in rats and mice on the relationship between periodontitis and rheumatoid arthritis shows a bidirectional interaction between these pathologies [38,39,40].

Periodontal bacteria such as *Aggregatibacter actinomycetemcomitans* and *Porphyromonas gingivalis* can induce autoimmune responses through the citrullination of host proteins, mediated by the pore-forming toxin Leukotoxin A and the enzyme peptidyl-arginine deiminase, respectively. These post-translational modifications may result in the production of rheumatoid arthritis-specific antibodies (ACPA), potentially exacerbating the disease in susceptible individuals [41]. Studies also suggest a link between periodontal disease and type 2 diabetes [42].

LPS could induce a systemic inflammatory reaction, possibly resulting in insulin resistance and the onset of type 2 diabetes. In vitro studies on animal models and cell cultures have indicated possible links between hyperglycemia and periodontal ligament cell function [43,44,45].

Elevated blood sugar levels due to hyperglycemia prompt the generation of significant amounts of glycation end products within periodontal tissues, initiating an inflammatory condition impacting cellular reactions. This inflammatory state of fibroblasts involves the release of increased levels of tumor necrosis factor-alpha, interleukin-6, and interleukin-1 beta [46,47].

LPS are involved in various pathologies, driving the search for strategies aimed at their prevention and treatment. Therapeutic approaches targeting LPS have been developed primarily in the context of sepsis, with the goal of neutralizing or eliminating LPS or inhibiting immune cell activation mediated by them to reduce the dysregulated host response. One such strategy involves using antibodies directed against LPS, such as E5 and HA-1A; however, these have shown limited effectiveness in sepsis treatment [29]. Another promising strategy is the development of neutralizing peptides, whereby protonated histidine-rich polypeptides can inhibit LPS-induced responses by binding to and neutralizing them. However, these peptides have exhibited moderate toxicity [48]. Additionally, other therapeutic approaches have been proposed for endotoxemia treatment, including the use of non-absorbable drugs that bind to LPS in the intestine, reducing its translocation and mitigating its harmful effects on the host [49]. These therapeutic strategies represent crucial areas of research in the quest for effective treatments against LPS-associated pathologies, with the potential to significantly enhance the clinical management of severe medical conditions.

## 4. Effect of Bacterial LPS on the Biological Characteristics of Dental Stem Cells

### Influence of LPS on Cell Viability and Cellular Differentiation

Dental issues such as dental caries, erosion, trauma, and bacterial infections can affect the vitality of dental tissues. There are reports in the literature regarding the effect of LPS on the biological characteristics of MSCs. Several studies have explored the impact of bacterial LPS on the adhesion, migration, proliferation, and differentiation of dental stem cells [50,51,52,53].

Cell proliferation and migration play crucial roles in tissue repair under both physiological and inflammatory conditions. Studies have shown that when dental pulp is damaged by bacteria, DPSCs can form reparative dentin as a protective barrier. Various approaches have been utilized to investigate the impact of different bacterial LPS on the adhesion, migration, proliferation, and differentiation of dental stem cells [50,51,52,53].

For instance, the use of LPS to simulate chronic exposure in a pulpitis setting has been explored with DPSCs. Mendes-Soares et al. (2023) found that treatment with 10 µg/mL of LPS reduced cell viability and odontogenic potential under pulpitis conditions [54]. However, Bindal et al. (2018) reported a decrease in viability when concentrations of 1 and 2 mg/mL of LPS were used in DPSCs [7]. Conversely, Widbiller et al. (2018) noted that treatment with LPS (0.1 µg/mL) did not affect viability but did interfere with odontoblastic differentiation in human DPSCs [55]. An approach to studying the response of dental stem cells to LPS involves the development of three-dimensional cultures with scaffolds.

For example, Firouzi et al. (2022) used a fibrin scaffold enriched with growth factors to assess the proliferation and differentiation of DPSCs under LPS-induced inflammatory conditions. They observed that DPSCs treated with LPS (1 µg/mL) on the fibrin scaffold promoted the release of pro-inflammatory cytokines such as TNF-α, which supported cell proliferation and migration during dental pulp inflammation [56].

Another interesting approach for treating pulpitis involves the use of milk-derived exosomes because they have been shown to reduce the inflammatory response induced by LPS and improve odontogenic differentiation [57]. Progress has been made in studies of scaffold-free 3D constructs formed by DPSCs, which involve culturing cells on dishes to create cell sheets as models of oral disease. Rothermund et al. (2022) reported that scaffold-free 3D constructs of DPSCs with LPS did not negatively impact their proliferation or apoptosis but did inhibit the positive regulation of RUNX2 mRNA and the expression of odontoblast-associated proteins [58].

Another significant disease with an inflammatory microenvironment induced by LPS is periodontitis. Studies have analyzed LPS in periodontal ligament stem cells (PDLSCs) [59,60] and apical papilla (SCAPs) [61]. In PDLSCs, it was found that treating cells with rutin, a flavonoid with anti-inflammatory properties, reversed LPS-induced damage through phosphatidylinositol 3-kinase (PI3K)/AKT signaling, promoting proliferation and osteogenic differentiation [59]. Some studies indicate that the osteogenesis potential is affected in PDLSCs at high LPS concentrations.

For example, Li et al. (2014) reported that 10 μg/mL of LPS negatively impacted osteogenic differentiation by inducing apoptosis through the NF-kB pathway [5]. Additionally, the effect of LPS on other types of dental stem cells, such as those derived from the apical papilla (SCAPs), has been studied. Lertchirakarn et al. (2017) found that 5 mg/mL of *Porphyromonas gingivalis* LPS altered the osteogenic differentiation of SCAPs by increasing the expression of the BSP gene [61]. Bucchi et al. (2024) reported that 1 μg/mL of *E. coli* LPS affected SCAPs viability, but they maintained their adhesion capacity to dentin and the mineralizing phenotype [62]. Ramenzoni et al. (2019) studied the immunomodulatory response of PLSCs and their commitment to cell differentiation using the supernatants of periodontal bacteria [63]. Transcriptional profile RNAseq analysis showed an increase in gene expression related to adipogenic or chondrogenic processes and an inhibition of osteogenesis [63].

DTSCs exhibit immunosuppressive activity that helps maintain homeostasis in dental tissues [15], preventing inflammatory responses. Therefore, researchers have sought molecules to improve the regenerative process of DPSCs in LPS-induced inflammation. Firouzi et al. (2022) observed that the use of concentrated growth factors promoted the release of cytokines such as TNF-α, which positively affected proliferation, migration, and odonto/osteogenic differentiation in DPSCs [56]. Another approach explored involves analyzing the behavior of DPSCs in interaction with bioactive materials to develop strategies in pulp therapy. Pedroza et al. (2022) reported that creating an inflammatory environment (1 μg/mL of LPS) in DPSCs using calcium silicate-based materials improved proliferation and mineralization potential while inducing anti-inflammatory mechanisms [64].

These findings suggest that the behavior of dental stem cells varies based on the origin of LPS (e.g., *E. coli* vs. *P. gingivalis*), type of dental stem cell, LPS concentration, and cytokine concentration released into the culture medium by LPS-exposed cells. Therefore, it is essential to conduct studies with clinically relevant bacterial LPS to determine the cellular and molecular mechanisms involved in the inflammatory process in dental stem cells and thereby develop novel regenerative strategies.

## 5. Molecular Players Involved in the Interaction between Dental Stem Cells and LPS

### 5.1. Molecular Sensors of LPS in Dental Stem Cells

Bacterial infection is the most common cause of diseases such as dental pulpitis and periodontitis in permanent teeth [7]. LPS are released through the lysis of bacteria or as vesicles from their membranes [65]. However, the molecular mechanism that occurs in the LPS-dental stem cell interaction is still not clear. Two suggestions have been put forward. The first is the recognition by the innate immune system of the lipid A fraction through the protein MD-2, which forms a complex with Toll-like receptors (TLRs; Figure 3) [5].

TLRs are type I transmembrane proteins composed of extracellular leucine-rich repeats (LRR), a single transmembrane domain, and a cytoplasmic Toll/IL-1 receptor (TIR) domain involved in the recruitment of signaling adapter molecules such as MD-2 [19,66,67]. MD-2 does not have a transmembrane domain, so it remains attached to the cell surface through its interaction with TLR4 [68]. TLRs are important mediators in inflammatory pathways and regulate migration or fate decisions in DTSCs [66,69]. After ligand binding, a cascade of signaling pathways, including cell type–dependent pathways through myeloid differentiation factor-88 (MyD88)-dependent, interferon b-dependent mitogen-activated protein pathway, TIR domain adapter kinase (MAPK)-inducing pathway, and phosphoinositide 3-kinase-Akt pathways, is triggered, which can influence proliferation, migration, and differentiation in MSCs and activate the expression of pro-inflammatory cytokines such as tumor necrosis factor-α (TNF-α), interleukin-6 (IL-6), and interleukin-8 (IL-8) [50,70]. To date, the differential expression of 10 members of TLRs has been identified in MSCs, including TLR4, whose ligand is LPS [70]. Lang et al. (2022) found that *E. coli* LPS, but not *P. gingivalis* LPS, promotes the expression of TLR4 and inflammatory cytokines in hDPSCs [6]. Liu et al. (2014) [69] demonstrated that the activation of TLR4 generated a positive regulation of migration in DPSCs but a negative effect on their proliferation.

The effect of TLRs on the differentiation of dental stem cells has been widely investigated. In hPDLSCs, TLR4 signaling is activated and expresses various inflammatory factors, including TNF-α and IL-1β, which inhibit osteogenic differentiation but induce adipogenesis [71]. Chen et al. (2022) reported that the activation of nuclear factor kappa B (NF-κB) signaling in an LPS-induced inflammatory environment can inhibit osteogenic differentiation [72]. With regard to DPSCs, Kim et al. (2023) reported that LPS treatment (1 μg/mL) improved odontogenic differentiation through C5aR and p38 signaling, which led to an increase in the expression of odontogenic lineage markers such as dentin sialophosphoprotein (DSPP) and dentin matrix protein 1 (DMP-1) [68]. Zhang et al. (2023) performed a single-cell sequencing analysis on DPSCs treated with LPS from two oral bacteria (*Porphyromonas gingivalis* and Enterococcus faecalis). They reported the stimulatory effect of *P. gingivalis* LPS on the mineralization of DPSCs.

Another signaling pathway that also participates in the response to LPS is Wnt5a. In this regard, He et al. (2014) [73] reported the expression of Wnt5a through the transduction of TLR4/MyD88, PI3-kinase/AKT, and NFkB signaling, which emphasizes their importance as molecular components during inflammation of the dental pulp. Another aspect that has been studied in DPSCs is LPS-induced senescence. In this context, Feng et al. (2013) reported that the production of ROS promotes the DNA damage response and expression of p16INK4A (mediators of the aging process), causing cell cycle arrest [74]. Therefore, the number of lipid A acyl chains affects the interaction between LPS and MD2 and the subsequent dimerization of TLR4-MD2. Hence, the role of lipid A is significant in the pro-inflammatory signaling cascade in which LPS elicits its toxicity.

### 5.2. Emerging Players in DTSCs-LPS

Several studies have shown that the response of DTSCs in bacterial inflammatory oral diseases is modulated by epigenetic mechanisms [75,76,77,78]. The most studied epigenetic mechanisms are DNA methylation, chemical modifications in histones, and non-coding RNAs. In this context, the role of long non-coding RNAs (LncRNAs) has been explored in the regulation of the regenerative capacity of DTSCs related to LPS-mediated inflammation. Han et al. (2019) [79] reported the function of LncRNA taurine-upregulated gene 1 (TUG1) in the inflammatory process induced by *Porphyromonas gingivalis*-derived LPS. The authors showed that TUG1 overexpression decreased LPS-stimulated proliferative inhibition in ligament cells, while its knockdown had the opposite effect [79].

Another interesting role of LncRNAs in inflammatory environments is the regulation they exert on microRNAs (miRNAs). miRNAs are recognized as molecules involved in the transcription of genes that regulate the inflammatory response in pulpitis and periodontitis [80,81]. Huang et al. (2020) showed that TUG1 could protect against periodontitis via regulating miR-498/RORA-mediated Wnt/β-catenin signaling [82]. Dong et al. (2020) demonstrated that overexpression of LncRNA maternally expressed gene 3 (MEG3) represses the LPS-induced response through the inhibition of the AKT/IKK and miR-143-3p pathways [83].

DNA methylation and histone acetylation are the main modifications that are induced in oral inflammatory diseases [78]. DNA methylation has been reported to modulate chromatin regions for the expression of immunoinflammatory genes related to periodontitis [84]. Mo et al. (2019) used next-generation sequencing to analyze whether miRNA expression could be regulated by DNA methylation in the inflammatory process generated by LPS from *Porphyromonas gingivalis* in DPSCs [85]. The authors demonstrated that inhibition of DNA methyltransferases affects the expression of miR-146a-5p, which is a miRNA related to pulp inflammation.

In terms of histone acetylation, it has been reported that histone deacetylases (HDACs) precisely regulate several physiological processes and diseases in DSCs [86]. Histone deacetylases (HDACs) have been observed to precisely regulate differentiation in DSCs. A previous study found that *HDAC1* and *HDAC3* transcripts are highly expressed in periodontitis. Diomede et al. (2017) [78] identified a differential regulatory effect of HDAC1, HDAC2, histone acetylase (p300), and DNA methyltransferase 1 (DNMT1) during the inflammatory process induced by LPS in hPDLSC. This highlights the role of key epigenetic players in periodontitis and pulp inflammation, potentially leading to the identification of biomarkers for therapeutic strategies in other oral diseases. Further research is needed to explore the role of epigenetic mechanisms in regulating the inflammatory response in dental stem cells.

## 6. Conclusions

DTSCs have the potential to be employed as a therapeutic source for the regeneration of dental and oral tissues damaged by inflammatory diseases. Several studies have shown that DTSCs can respond to LPS stimuli by modulating their behavior. However, before their clinical application in patients, some challenges need to be addressed. To overcome these challenges, research could focus on three directions. The first is to investigate the molecular mechanisms underlying the therapeutic behavior of DTSCs during inflammation. This includes the analysis of paracrine factors, target molecules, and transcription factors. The second is to develop three-dimensional culture technologies to study the biological behavior of DTSCs in in vivo environments such as the inflammatory process generated by LPS-producing bacteria. This is a complex and crucial direction, as it will provide valuable information on how DTSCs behave in real conditions. The latter involves controlling the molecular program of DTSCs to ensure that they carry out their therapeutic effects accurately in vivo. This could involve techniques such as genetic engineering or small molecule inhibitors/agonists to regulate specific signaling pathways and optimize their responses. Advances in these directions will be essential for the future application of DTSCs in the treatment of diseases caused by LPS.

## Figures and Tables

**Figure 1 ijms-25-04338-f001:**
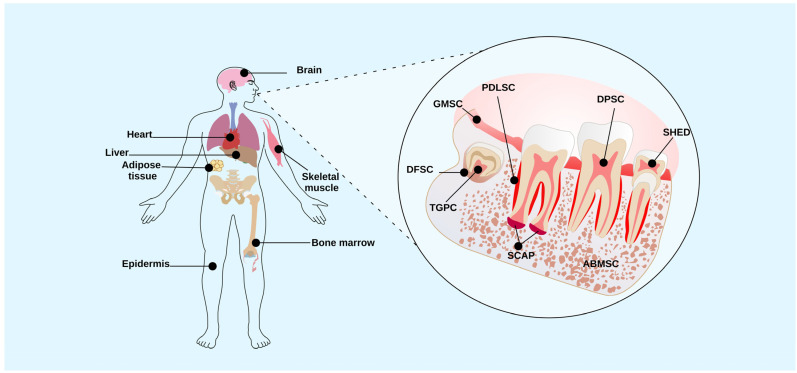
Adult stem cell niches. Adult stem cells are in different niches of the human body, such as the heart, liver, brain, adipose tissue, bone marrow, epidermis, skeletal muscle, and oral cavity. In the oral cavity, there are populations of stem cells derived from different dental regions such as DFSC, dental follicle stem cells; TGPC, tooth germ progenitor cells; GMSC, gingiva-derived mesenchymal stem cells; SCAP, stem cells from apical papilla; PDLSC, periodontal ligament stem cells; DPSC, dental pulp stem cells; ABMSC, alveolar bone–derived mesenchymal stem cell; and SHED, stem cells from human exfoliated deciduous teeth.

**Figure 2 ijms-25-04338-f002:**
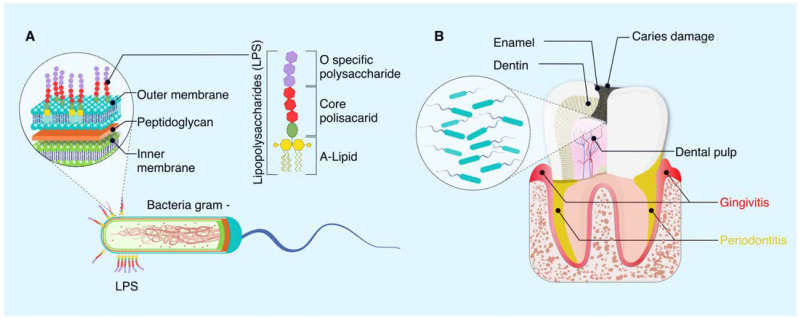
Schematic diagram of the chemical structure of LPS and their influence on oral diseases. (**A**) LPS are found in the outer membrane of gram-negative bacteria; their structure is composed of a hydrophobic part, known as A-Lipid, inserted in the lipid bilayer of the outer membrane, which is responsible for toxicity. This in turn has a polysaccharide core made up of heptoses and hexoses attached to A-Lipid. On the external surface of the structure, the specific O polysaccharide is attached, which is made up of carbohydrates that are variable depending on the bacterial species (**B**) LPS contribute to the development of oral diseases, such as gingivitis, periodontitis, and pulpitis.

**Figure 3 ijms-25-04338-f003:**
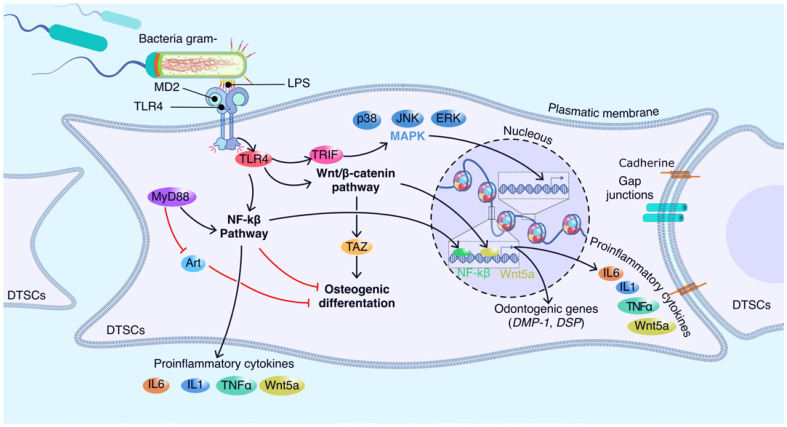
Molecular mechanisms of LPS bacterial interaction with dental stem cells. Binding of LPS induces dimerization of two TLR4-MD2 units, and dimerization of the TLR4-MD2 complex signals physiological recognition of LPS. The lipid A component is responsible for LPS binding to MD2. The resulting complex then binds to TLR4 and triggers a signaling cascade leading to the production and secretion of pro-inflammatory cytokines that activate to NF-kβ pathway, Wnt/β-catenin pathway, and MAPK. These events accumulate pro-inflammatory signaling in DTSCs and surrounding causes, and decreased differentiation. The NF-kβ signaling pathway may have a dual effect on DTSCs, they may inhibit osteogenic differentiation or induce the activation of genes involved in odontogenic differentiation such as DMP-1 and DSP. On the other hand, the Wnt/β-catenin signaling pathway promotes osteogenic differentiation and the expression of odontogenic genes. In both pathways, the production of pro-inflammatory cytokines, such as IL-6, IL-1, TNF-α, and Wnt5a, is triggered. DTSCs, dental tissue-derived mesenchymal stem cells; LPS, lipopolysaccharide; MAPK, mitogen-activated protein kinase; MyD88, myeloid differentiation factor 88; NF-kβ, nuclear factor kappa beta; PM, plasma membrane; receptor activator of NF-kβ; RNKL, Receptor activator of NF-kβligand; TGF-β2, transforming growth factor beta2; TLR, toll-like receptor; Wnt5a, Wnt Family Member 5A.

## Data Availability

Not applicable.

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
