# Peer review of "Dental Stem Cells and Lipopolysaccharides: A Concise Review"

_ijms, 2024, doi:10.3390/ijms25084338_

Round 1

Reviewer 1 Report

Comments and Suggestions for Authors

This review explores the interaction between dental tissue-derived mesenchymal stem cells (DT-MSCs) and lipopolysaccharides (LPS), particularly focusing on the effect of LPS on the biological behavior of DT-MSCs and the molecular mechanisms involved in this interaction.

The abstract introduces the significance of DT-MSCs in regenerative medicine and their role in the immune response to oral inflammatory diseases triggered by LPS. It highlights the recognition of LPS by Toll-like receptors (TLRs) in dental stem cells and discusses the need for further exploration of their clinical application.

  1. Provides an overview of the oral microbiome and the inflammatory diseases caused by bacteria, emphasizing the impact on DT-MSCs.
  2. Describe the characteristics and potential applications of DT-MSCs in regenerative medicine.
  3. Explores the structure of LPS and its role in various oral diseases, including periodontitis, gingivitis, and dental caries.
  4. Discusses the influence of LPS on cell viability, proliferation, migration, and differentiation of dental stem cells, including dental pulp stem cells (DPSCs), periodontal ligament stem cells (PDLSCs), and stem cells from the apical papilla (SCAP).
  5. Examines the molecular mechanisms underlying the interaction between LPS and dental stem cells, including the role of Toll-like receptors (TLRs), NF-κB signaling, Wnt/β-catenin pathway, and the impact of epigenetic factors such as long non-coding RNAs (LncRNAs).
  6. Discusses recent research on the role of LncRNAs in regulating the inflammatory response of dental stem cells to LPS.
Comments on the Quality of English Language

Nil

Reviewer 2 Report

Comments and Suggestions for Authors

This is a short review of the potential biological connection between LPS and dental MSCs. I have following comments: 

1. The title is not correct. Dental stem cells are not MSCs. 

2. What's DTSCs of line 66 and Figure 2? 

3. Fig.1 and 2 are easy for understanding. However, you need a figure to display the exact location of different dental stem cells, dental MSCs, and bone marrow MSCs. 

4. Section 5.2, you need to correlate the new players to MSCs, in addition to their connection to LPS.

5. What's the challenge (s) of line 368? Also, it's confusing to read your conclusions. What's the clinical implications, in terms of pathology or treatments?

Comments on the Quality of English Language

English is fine, but the knowledge needs to be improved. Also, need to think about what message you really want to deliver. 

Round 2

Reviewer 2 Report

Comments and Suggestions for Authors

Thank you for the revision.